# Pilot protocol for the Parent and Infant Inter (X)action Intervention (PIXI) feasibility study

**Anne C. Wheeler**[1]*, **Katherine C. Okoniewski**[1], **Samantha Scott**[1], **Anne Edwards**[1], **Emily Cheves**[1], **Lauren Turner-Brown**[2]

**1** RTI International, Research Triangle Park, North Carolina, United States of America, **2** Department of Psychiatry, University of North Carolina at Chapel Hill, Chapel Hill, North Carolina, United States of America

\* acwheeler@rti.org

## Abstract

This paper provides the detailed protocol for a pilot study testing the feasibility, acceptability, and initial efficacy of a targeted two-phase, remotely delivered early intervention program for infants with neurogenetic conditions (NGC) and their caregivers. The Parent and Infant Inter(X)action Intervention (PIXI) is designed to support parents and infants with a NGC diagnosed in the first year of life. PIXI is implemented in two phases, with the first phase focusing on psychoeducation, parent support, and how to establish routines for supporting infant development. Phase II helps parents learn targeted skills to support their infant's development as symptoms may begin to emerge. The proposed non-randomized feasibility pilot study will establish the feasibility of a year-long virtually implemented intervention program to support new parents of an infant diagnosed with an NGC.

**Data Availability Statement:** No datasets were generated or analysed during the current study. All relevant data from this study will be made available upon study completion.

## Introduction

The last decade has witnessed an expansion of prenatal carrier testing [1] and newborn screening studies [2] designed to address the diagnostic odyssey frequently experienced by families affected by a neurogenetic condition (NGC). NGCs are disorders caused by a genetic defect that results in alterations to brain function, with an increased potential for neurodevelopmental and intellectual disabilities. Examples of NGCs include fragile X syndrome (FXS), Angelman syndrome (AS), Prader-Willi syndrome (PWS), Dup15q syndrome (Dup15q), and Williams syndrome (WS), among many others. While the range of genetic conditions and accompanying needs are wide and vast, for the purposes of the proposed intervention program, the NGCs included are exclusive to those with a primary phenotypic feature of developmental delay and/or intellectual disability.

Reducing the time from first concern to diagnosis is important for ensuring young children with NGCs have access to early intervention (EI) and/or disease modifying therapeutics as early as possible. With these new initiatives to reduce the age of diagnosis, newborns with NGCs will be diagnosed long before parents or pediatricians would have otherwise been concerned for the child's development. Early intervention in children who are pre-symptomatic primarily involves surveillance, with the onset of direct intervention only occurring after delays have been documented. This approach is practical and resource effective; however,

**Funding:** AW John Merck Fund The funders had and will not have a role in study design, data collection and analysis, decision to publish, or preparation of the manuscript.

**Competing interests:** The authors have declared that no competing interests exist.

intervening *before* symptoms have emerged in children with known NGCs with predictable sequalae, offers an enhanced opportunity to potentially reduce the long-term impact of those symptoms. By definition, NGCs result in changes in brain function that cannot be undone through behavioral interventions. As demonstrated by work being done with infant siblings of children with a known autism spectrum disorder (ASD), parent-mediated interventions that teach parents how to recognize and address their child's unique developmental profiles can help reduce or ameliorate the downstream neurological impact of their condition [3] and improve quality of life for the individual and the family.

Although NGCs as a group are highly diverse regarding developmental and behavioral phenotypes, there are some core developmental domains that can be prioritized that will theoretically be beneficial for all. These include interventions that target *Early Communication* (including social communication), *Visual Attention*, and *Motor Skills*. Focusing on strategies to improve communication, motor skills, and visual attention has the potential for a cumulative effect on developmental trajectories with a downstream effect on the overall cognitive and behavioral repertoire of children with NGCs as they get older. For example, a focus on motor coordination aims for improvement in postural strength [4], which in turn can provide opportunities for increased attention to surroundings. Similarly, teaching parents to get in their infant's line of sight when interacting may improve attention and communication, providing potential compensation for delays in visual tracking and motor coordination.

Several other components are important to consider in the development of an intervention program for families affected by NGCs. First, decades of research on efficacy of EI practices point to strategies that can be implemented by parents within daily routines as having the most sustainable impact on child development [5]. Therefore, intervention procedures that embed a parent-mediated, routines-based approach are likely to be most effective. Second, although high-intensity intervention models (e.g., 20–30 hours per week) can be effective in improving cognitive outcomes for young children at familial risk for autism spectrum disorders [6], it is unlikely to be feasible or sustainable for most families. Therefore, an effective intervention that requires moderate intensity is more desirable. Third, given the rare nature of NGCs, an intervention that could be implemented remotely and therefore could be accessible to a larger proportion of the population is likely to have a wider reach, improving access. Lastly, when targeting infants in the first year of life after receiving a diagnosis of a NGCs, an intervention model needs to incorporate challenges that may arise for parents coping with postpartum changes and a steep learning curve to understand their child's condition.

To date, there are no known studies of targeted interventions for infants with NGCs in the first year of life; therefore, a functional review of existing intervention models focused on the aforementioned key components was employed to identify empirically based programs that could set the foundation for a targeted intervention program for infants with NGCs and their families. Our criteria for review included intervention models that were (1) conducted with infants under 12 months of age; (2) targeted at least two of our three identified developmental domains (i.e., communication, motor, visual attention); (3) parent-mediated (4) based on empirical evidence of efficacy as documented by at least one randomized clinical trial and (5) included adaptations for telehealth delivery or deemed feasible for remote administration. Based on this review, we identified two sources on which to base the core of the intervention program: Parents as Teachers (PAT) [7] and content adapted from Infant Start manual [8].

The current paper outlines the protocol for *Parents and Infants Inter(X)action Intervention* (PIXI), a non-randomized feasibility pilot trial of a remotely delivered, two-phase, EI program targeting emerging symptoms in infants with NGCs. A mixed-methods approach to exploring intervention acceptability, feasibility, and initial efficacy was chosen to assure comprehensive exploration of outcomes that include the voices of affected parents. These methodologies

include quantitative and qualitative measures to learn more about the early experience of infants diagnosed with NGCs and their families within the first year of life, intervention acceptability, and potential response to intervention. Outcomes of the feasibility trial will inform the development of a powered randomized controlled trial to determine the efficacy of PIXI for infants with NGCs. The protocol is described in accordance with the Standard Protocol Items: Recommendations for Interventional Trials (SPIRIT) guidelines [9, 10] with CONSORT extension clarifications for pilot trials [11].

## Objectives

The primary objectives of this study are to assess the feasibility and acceptability of a virtually delivered targeted EI program for infants with NGCs and their caregivers, to inform iterative changes to the intervention itself, and to determine utility of outcome measures in preparation for a randomized control trial. To meet this objective, the following components will be assessed as primary feasibility and acceptability outcomes:

1. Recruitment rates and obstacles to enrollment, including technology and schedule needs

2. Retention of families through the full intervention timeline

3. Acceptability of the intervention and the various components, as determined by parent responses on self-report questionnaires and qualitative interviews

4. Parent and therapist fidelity of intervention administration

Our secondary objective is to assess initial the applicability, feasibility, and validity of child and family outcome measures for future efficacy trials. We will assess the utility of measures designed to assess child development, co-occurring behavioral issues or conditions (e.g., Autism, ADHD), parental well-being and parental efficacy.

## Design

This will be a single-arm non-randomized feasibility pilot feasibility study. A minimum of 20 and up to 30 infants with one of five initially targeted NGCs (FXS, AS, PWS, Dup15q, WS) and their families will participate in this intervention program over the course of the infants' first year of life. Assessments of the infants and family well-being will be conducted three times over the course of the first year—pre-intervention, mid-way through the intervention, and post-intervention—and annually through the child's third birthday (see Fig 1).

## Methods

### Study setting

Our target NGCs are all rare neurogenetic disorders. To accommodate families across a large geographical region, PIXI was designed to be fully implemented remotely. Families who do not have access to required technology will be sent a Microsoft Surface Go 8GB Ram and 128GB SSD, one Surface Go case, one Logitech C922 Pro Stream Webcam 1080P camera, and one USBC-to-USB adapter to ensure technological access. If participants do not have reliable internet service, they will be sent a wireless hotspot or booster, depending on need. A technology session will be offered to families at the beginning of the intervention to ensure a reliable connection and comfort with the tools and programs prior to the start of the sessions.

The interventionist will connect with the family from a secure, private location to assure confidentiality throughout the sessions. Recordings will be made for the purpose of fidelity assessments and behavioral coding of the infant and parent.

| | Study Period | | | | | | | | | | |
| | Enrollment | Intervention | | | | | | | | | Long-Term Follow-Up |
| **TIMEPOINT*** | -T 1 | | T 1 | - | T 2 | - | T 3 | T 4 | T 5 | T 6 | T 7 |
| **Enrollment** | | | | | | | | | | | |
| Eligibility Screen | x | | | | | | | | | | |
| Informed Consent | x | | | | | | | | | | |
| Allocation | | | | | | | | | | | |
| **Interventions** | | | | x | | x | | | | | |
| Phase I | | | | x | | | | | | | |
| Phase II | | | | | x | | | | | | |
| ASSESSMENTS | | | Pre-Intervention 1 | | Mid-Way Point 2 | | Post-Intervention 3 | | | | |
| **Baseline Variables** | Length (mins) | | | | | | | | | | |
| Current Concerns & Interventions | 5 | | x | | x | | x | x | x | x | x |
| Vineland-3 | 10 | | x | | x | | x | x | x | x | x |
| **Primary Outcome Variables** | | | | | | | | | | | |
| Social Validity | 10 | | x | | x | | | | | | |
| Parent Interview | 20 | | | | x | | | | | | |
| **Secondary Outcome Variables** | | | | | | | | | | | |
| Infant Start Parent Fidelity | 5 | | | x | | | | | | | |
| ASD Clinical Evaluation | 60 | | | | | | x | x | | | |
| ***Parent Report: Child*** | | | | | | | | | | | |
| Feeding Flock | 30 | x | x | | x | | x | x | x | x | x |
| BISQ Adapted | 5 | x | x | | x | | x | x | x | x | x |
| IBQ-R VSF | 30 | x | x | | x | | x | x | x | x | x |
| CSBS-DP | 20 | | x | | x | | x | x | x | x | x |
| MCHAT | 10 | | | | | | | x | | | |
| SP-2 | 20 | x | x | | x | | x | x | x | x | x |
| Repetitive Behavior Scale | 15 | | | | x | | | x | x | x | x |
| ***Parent Report: Self*** | | | | | | | | | | | |
| Demographics | 5 | x | x | | x | | x | x | x | x | x |
| Edinburgh | 5 | x | x | | x | | | | | | |
| PHQ-9 | 5 | | | | | | | x | x | x | x |
| BRIEF2A | 15 | | | | x | | | | | | |
| STAI | 20 | | x | | x | | x | x | x | x | x |
| PSI-4: SF | 10 | | x | | x | | x | x | x | x | x |
| MOS-SSS | 5 | | x | | x | | x | x | x | x | x |

Vineland-3, Vineland Adaptive Behavior Scales, Third Edition; ADOS-2, Autism Diagnostic Observation Schedule, Second Edition; BISQ Adapted, Brief Infant Sleep Questionnaire; IBQ-R VSF, Infant Behavior Questionnaire Revised Very Short Form; CSBS-DP, Communication and Symbolic Behavior Scales-Developmental Profile; MCHAT, Modified Checklist for Autism in Toddlers; SP-2, Sensory Profile, Second Edition; Edinburgh, Edinburgh Postnatal Depression Scale; PHQ-9, Patient Health Questionnaire-9;

**Fig 1. SPIRIT schedule of enrollment, interventions and assessments for a pilot trial.** Vineland-3, Vineland Adaptive Behavior Scales, Third Edition; ADOS-2, Autism Diagnostic Observation Schedule, Second Edition; BISQ Adapted, Brief Infant Sleep Questionnaire; IBQ-R VSF, Infant Behavior Questionnaire Revised Very Short Form; CSBS-DP, Communication and Symbolic Behavior Scales-Developmental Profile; MCHAT, Modified Checklist for Autism in Toddlers; SP-2, Sensory Profile, Second Edition; Edinburgh, Edinburgh Postnatal Depression Scale; PHQ-9, Patient Health Questionnaire-9; BRIEF2A, Behavior Rating Inventory of Executive Functioning, Second Edition; STAI, State-Trait Anxiety Inventory; PSI-4: SF, Parenting Stress Index, Fourth Edition; MOS-SSS, Medical Outcomes Study-Social Support Scale. *Time 1 (Baseline, pre-Phase I), Time 2 (6–9 months of age; post-Phase I/pre-Phase II), Time 3 (12–15 months of age; post-Phase II), Time 4 (18 months of age), Time 5 (24 months of age), Time 6 (30 months of age), Time 7 (36 months of age).

## Participants and sample size

We will aim to recruit 20–30 families over a 4-year period. Eligible participants must meet the following criteria: (1) be under 12 months of age with a confirmed diagnosis of one of the target NGCs; (2) have received a diagnosis of one of the target NGCs prior to emergence of symptomology (i.e., diagnosis was not solely due to parental concerns about the infant's development); and (3) be English-speaking. Participants will not be excluded based on race, ethnicity, or biological sex; however, English must be the primary language spoken at home as all assessment measures and intervention protocols for the current pilot protocol are in English. Either one or both parents are eligible to participate in PIXI programming.

## Recruitment and enrollment procedures

Families will be recruited through multiple sources: the primary source will be infants identified with one of the target NGCs through Early Check, a voluntary newborn screening study conducted in North Carolina [12]. Participants will also be recruited through state and national patient advocacy groups via email listservs and social media postings. A genetic report will be required to verify diagnosis.

Following confirmation of eligibility and receipt of consent, a baseline assessment will be conducted (see Fig 2). A remote developmental assessment battery [13] will be administered to all participants along with parent questionnaires that assess infant development, feeding, eating, sleeping, and temperament. Parent well-being (e.g., depression, anxiety, stress) will also be assessed at each time point (see *Outcome Measures* below).

In addition to any needed equipment for participation in virtual sessions, enrolled participants will also receive all toys and supplies needed for activities: one set of stacking blocks, a blanket, seven board books, lotion, rattle, bubbles, 12-inch ball, and collapsible bin for storage.

## Interventionists

Study interventionists will be licensed clinicians (e.g., licensed clinical social workers, licensed psychologists) with expertise working with young children and their parents. Interventionists will complete the 30-hour foundational training course to implement PAT materials [7]. A 2-day training on the manualized content of Infant Start [8], parent coaching strategies, and development of individualized goals will be led by the co-developer of Infant Start, Laurie Vismara, with ongoing consultation to the PIXI interventionists, following the initial training. After training completion, interventionists will continue to participate in peer supervision and didactic exchanges about adaptations and its use to ensure coordinated implementation across participants. Visit records and intervention fidelity measures will be completed during every session to ensure each component is executed in accordance with programming guidelines. Prior to implementation of Phase 2, the interventionists will meet ESDM Parent Coaching Fidelity [14] requirements as defined by a score of 80% or greater across three consecutive sessions with two separate participants. Inter-rater reliability will be established among three

**Enrollment**
- Infant identified with NGC prior to 12-months of age and emergence of symptomology
- Baseline assessment of developmental skills, infant behavior, and parent concerns

**Phase 1**
- Parents as Teachers materials and curriculum
- Four to eight sessions over 8–12 weeks
- Time 2 assessment at completion of Phase 1 (6–9 months)

**Phase 2**
- Adapted Infant Start programming
- 12-14 sessions between 6 and 15 months of age
- Time 3 assessment at completion of Phase 2 (12–15 months)

**Longitudinal Follow-Up**
- Time 4, 5, 6, and 7 assessments coordinated with 18-, 24-, 30-, and 36-months of age
- Outcome measures include social validity, parent feasibility, early developmental skills, and autism symptomology

**Fig 2. PIXI participation flow.**

coders during first two participants. Coaching fidelity will be maintained throughout the study by self-ratings immediately after the sessions and one coder independently rating 20% of all sessions administered.

## Cost monitoring

Across the trial cost for implementation will be closely recorded and considered for future scalability. Records of interventionist time for training, fidelity attainment, planning, intervention execution, and supervision will be made along with tracking of costs for materials and trainings.

## Intervention

A brief overview of the intervention is provided in Table 1. PIXI will involve two phases: Phase I will incorporate the PAT curriculum and will focus on building rapport and providing psychoeducation for families of newly diagnosed infants; Phase II will be based on the Infant Start framework and will involve active practice of skills by parents in interactions with their infants. These programs will serve as the foundation and provide structure for the sessions; consideration for adaptations based on needs of the infants and families will be made through iterative discussions with the study team.

**Table 1. Brief overview of the intervention.**

*PIXI Phase 1- Modified from Parents as Teachers Curriculum*

| Session | Parent-Child Interaction | Development-Centered Parenting | Family Well-Being |
|---------|--------------------------|-------------------------------|-------------------|
| Session 1 | Play is Learning | Introduction: Development Centered Topics | Introduction: Well-Being Topics |
| Session 2 | Creating Lifelong Learners- Book Routines/Language Development | Routines | Recreation and Enrichment |
| Session 3 | Social Development | Attachment and Social Development | Mental Health and Wellness |
| Session 4 | Cognitive Development | Physical Health | Physical Health—Family |
| Session 5 | Development through Senses | Nutrition | Basic Essentials |
| Session 6 | Motor Development | Sleep | Social Support |
| Session 7 | Motor Development | Discipline | Mental Health—Coping with Stress |
| Session 8 | Motor Development—Building Strength | Transitions | Early Care and Education |

*PIXI Phase 2 –Modified from Infant Start curriculum*

| |
|---|
| Session 1: Setting the Stage |
| Session 2: Joining the Spotlight |
| Session 3: Follow the Lead |
| Session 4: Imitation |
| Session 5: Turning on Your Baby's Voice |
| Session 6: Expanding Your Baby's Sounds |
| Session 7: Moving Bodies**Session content not in original Infant Start curriculum |
| Session 8: Sharing Toy Play with Others |
| Session 9: Talking Bodies and Joint Attention |
| Session 10: Flexibility in Play *Complete second on more targeted repetitive behavior if present |
| Session 11: Communication Next Step |
| Session 12: Next Steps |

**PIXI intervention—Phase I.** Phase I will begin when the infant is between 3 and 7 months old, depending on the age of diagnosis and enrollment in the study. The intention is that all participants transition from Phase I to Phase II by the time the infant is 9 months of age. The first phase of PIXI aims to (1) establish rapport with the family and infant and develop a rhythm of scheduled sessions; (2) engage in parent-child interactive activities, setting the foundation for work covered during Phase II; and (3) provide support and psychoeducation to parents and families about their child's diagnosed NGC and early child development. Sessions will be offered weekly or biweekly (frequency depending on the age of the infant and needs of the family), with a total of eight sessions administered. Phase I is designed to be flexible and be adapted based on the family or infant needs. However, each session will follow a structured format with guiding topics from the PAT curriculum. The format for Phase I sessions can be found in Table 2.

PAT [7] is an evidence-based program based in human ecological and family systems theory that provides services within the family context focused on parent-child interaction and relationships. Initially developed to enhance knowledge around child development and positive parenting skills, PAT uses parent coaching and empowerment strategies in a variety of parent populations ranging from birth to school age with a focus on both parent- and child-level outcomes. PAT sessions include three primary components designed to provide support around enhancing family protective factors and parent empowerment: parent-child interaction, development-centered parenting, and family well-being. To accommodate unique challenges that may arise from new parents facing a potentially unexpected diagnosis, we will embed a psychoeducational component that includes opportunities for parents to gain additional knowledge about their child's NGC and process their emotions about the diagnosis.

**Table 2. Sample Phase I session.**

| Component | Description |
| --- | --- |
| Weekly Check-in | • Events from the past 1–2 weeks<br>• Concerns/questions that arose<br>• How did the family continue to use the information provided in the last session? |
| Doing Anything New? | • Opportunity for parents to share observed skills in the past 1–2 weeks<br>  ○ Feeding, sleep, play, motor, language |
| Introduction to Session | • Interventionist shares with parent(s) what the topic of the session is |
| PAT Parent-Child Interaction | • A standard parent-child interaction activity is presented and completed with parent(s)<br>• Reflection: What did you notice about baby? What did you notice about yourself?<br>• Review of any additional parent handouts pertinent to the activity completed |
| PAT Developmental-Centered Parenting | • Introduces developmental topic<br>• Review of parent handout specific to area of development |
| PAT Family Well-Being | • Introduction of category/resources being addressed<br>• Review of parent handout specific to family well-being topic |
| Summary and Reflection | • Which part of today was most valuable?<br>• Questions |
| Next Session | • Set date and time |

*PAT: Parents as Teachers; Foundational Curriculum

1. *Parent-Child Interaction*: The goal of parent-child interaction discussion and activities is to provide the opportunity for parents to learn how to interactively engage with their infant through a facilitated assisted-learning model. Parents are provided with activity sheets and handouts that provide context and guidance about the activity as the interventionist reviews and provides real-time affirmation of efforts. After the activity, parents are asked to reflect on things they noticed their baby doing and things they noticed in themselves. Phase I activities were selected from Foundational Curriculum materials that allowed for free play, sensory play, and motor activities.

2. *Development-Centered Parenting*: The PAT Foundational Curriculum includes a discussion of seven developmental topics: attachment, discipline, health, nutrition, safety, sleep, and transitions and routines. The goal is to have parents better understand developmental progress and adapt their parenting behaviors to meet their child's needs at a specific stage in development. A review of all topics is included in Phase I, with flexibility around the discussion dependent on parent questions, family routines, and familiarity with concepts.

3. *Family Well-Being*: Within the PAT model, family well-being aims to bring to light the impact of parent and family circumstance and experience along with constant familial growth and change, on development, early learning, and overall health. Family well-being topics include basic essentials (e.g., food, nutrition, housing), education and employment (e.g., job training, college), physical health (e.g., medical and dental, injury prevention), mental health and wellness (e.g., stress, crisis intervention, community wellness services), early-care and education (e.g., childcare, preschool), relationships with family and friends (e.g., social support networks, community groups), and recreation and environment (e.g., community outings, leisure opportunities). All family well-being topics are presented to Phase I participants.

4. *Psychoeducation about the NGC.* Phase I includes provision of educational materials about their child's NGC based on needs and questions raised by each participating family. Materials will be developed in conjunction with EI service providers with expertise in each target NGC, licensed genetic counselors, family advocates, and advocacy organizations (e.g., the National Fragile X Foundation). These resources include information about the basic genetics of the condition and generational family implications (if applicable); information for providers and physicians about the disorder; early child development topics such as feeding, eating, sleeping, and child development. Eligibility for community-based EI programming will be discussed with parents during this phase, and referrals made to local EI providers as warranted. With parent permission, PIXI interventionists will work closely with EI service providers to enhance communication and coordinated care across settings.

After completion of each session, the interventionist will complete a modified version of the PAT visit record to ensure consistent reporting of topics covered during each session, aligned with PAT fidelity, and the opportunity to integrate clinical notes used for iterative development and qualitative analysis of needs. See Table 2 for an outline of session flow and components.

**Phase I to Phase II transition.** Upon completion of Phase I, a transition session will be completed between participants and interventionists. This transition session focuses primarily on the differences in procedures and protocols between the two phases, including session opening play sessions, topic review, parent coaching, and discussion of week-to-week practice topics. Interventionists will also take time during this session to orient to family priorities and current concerns, and answer questions. A 10-minute recorded parent-child free play interaction will also be conducted as a baseline assessment prior to Phase II. At the conclusion, a schedule for Phase II is discussed.

**Phase II.** Phase II will adapt Infant Start [8], a downward extension of the Early Start Denver Model (ESDM) curriculum [15], as an individualized, low-intensity parent-delivered intervention targeted at reducing and altering six developmental target symptoms and developmental patterns of ASD. Infant Start attempts to meet the unique needs of young infants suspected and later diagnosed on the autism spectrum by implementing practices identified in efficacious interventions for infants, including (1) parent coaching with a focus on both parental responsivity and sensitivity to child cues and teaching families to implement infant interventions; (2) individualization to each infant's developmental profile; (3) a focus on a broad rather than a narrow range of learning targets, and (4) interventions beginning as early as possible and providing greater intensity and duration of the intervention [16]. Infant Start is a manualized, unpublished intervention in which parents are coached on techniques to address six target symptoms: (1) visual fixations on objects; (2) abnormal repetitive behaviors; (3) lack of intentional communicative acts; (4) lack of coordination of gaze, affect, and voice in reciprocal turn-taking interactions; (5) lack of age-appropriate phonemic development; and (6) decreasing gaze, social interest, and engagement. A pilot study of Infant Start found significantly fewer symptoms and lower rates of ASD and developmental quotient scores under 70 at age 36 months in participating infants than a similarly symptomatic group of infants not enrolled in the study [16]. The model was well received by parents, and parents were able to achieve high fidelity of implementation at a low intensity (12 weekly sessions).

In addition to the core themes and manualized chapters of the original Infant Start manual, an additional chapter focused on motor development has been developed to meet the unique needs of infants with NGCs. This chapter, developed in collaboration with one of the Infant Start co-developers and infant physical and occupational therapists, focuses on supporting parental observation and understanding of infant motor development along with strategies to foster strength building and increased functional movements of the baby's hands and body.

PIXI child participants will begin Phase II by 9 months of age. Phase II consists of 12 weekly or bi-weekly, 1-hour telehealth sessions. Based on completed milestone checklists and discussion with parents about current concerns and goals for intervention, the interventionist develops three to six learning objectives to target during treatment. Learning objectives are based on each infant's developmental profile of strengths and areas of weakness, with focus in domains that are related to social learning and social-cognitive development, including verbal and nonverbal language, social orienting, and joint attention. Learning objectives also include a focus on motor development, as this is often one of the first symptoms reported by parents of infants with the target NGCs [17], and delays in motor skills often lead to delays in communication. Learning objectives are targeted during each session and embedded into the practice and reflection activities.

Prior to each session, the interventionist will send the family a corresponding coaching chapter, capturing a summary of the topic and written explanation of strategies and scenarios to be the focus of that upcoming session. Each session consists of six sequenced 5- to 10-minute activities. Sessions begin with a *greeting/check-in* in which the interventionist gathers information about implementation of strategies and child progress throughout the time between sessions, followed by a *warm-up activity* between child and parent to demonstrate previously learned skills and provide a reflection opportunity with interventionist on goals and elicited child behaviors. The interventionist then *introduces topic and theme* for the session and provides verbal and written information to parent about manualized session content (see Table 3). The parent then *practices* using the strategies while the interventionist uses coaching strategies to support the parent's implementation and self-reflection. The parent is then encouraged to *practice the strategy within another play or caregiving activity* (e.g., feeding, dressing, books, social play), while the interventionist again coaches the parent in implementation and self-reflection. The session *closing* involves a discussion between the parent and interventionist about how to generalize these strategies across settings, routines, materials, and family members (see Table 3).

The interventionist uses parent-coaching strategies [18] to help parents use the interactive strategies with their infant. The first six sessions of Phase II focus on setting the stage for parent-child interactions, getting into the spotlight, following the infant's lead, and supporting

**Table 3. Sample Phase II session.**

| Component | Description |
|---|---|
| Greeting/check-in | • Parent provides overview of week<br>• Coach acknowledges parent use of knowledge or skills during time between sessions<br>• Parent reflects on success/difficulties of implementation and any changes noted in child's behavior |
| Warm-up | • Un-interrupted parent-child activity that allows coach to gauge parent's understanding and implementation of strategies discussed and practiced in last session<br>• Coach shares reflections to reinforce parental use of techniques |
| Introduction to topic | • Sets the main topic for the session<br>• Information provided in multimodal teaching strategies (visuals, printed materials, verbal explanation, verbal demonstration)<br>• Coach emphasizes relationship among target child behaviors, main parenting strategies, and effects on child behavior |
| Coaching/ Reflection | • Parent practices use of new technique while coach provides enough support to help parent and child succeed<br>• Coach supports mastery and generalization using multiple activities<br>• Coach encourages parent to reflect on goals of the activity and the child's behavior |
| Closing | • Coach elicits parent's understanding of the topic<br>• Coach supports parent to develop a clear plan to use techniques throughout the week |

early language development. After session 6, the family participates in a mid-point check-in for the family to provide updates on family status, ask questions about the NGC or Phase I topics (e.g., development-centered parenting, family well-being), and follow up on any coordination of community-based EI services. The remaining six sessions focus on motor development, restricted and repetitive play and behaviors, joint attention, and bundling social communication skills. After completing all 12 sessions, families collaborate with the interventionist to identify a topic for a booster session in which previous sessions are repeated. Determination of topic may be based on parents' goals and/or interventionist's evaluation of parent's implementation of strategies. A minimum of one booster session is completed, with a maximum of three booster sessions delivered bi-weekly until the child turns 15 months of age.

The interventionist fidelity of treatment implementation is assessed and measured using the ESDM Parent(s) Coaching Fidelity system [14], a 5-point rating scale of 13 coaching behaviors: (1) review of progress; (2) demonstration of parent implementation; (3) introduction of topic; (4) coaching on topic; (5) coaching on generalization of topic; (6) discussion of generalization; (7) collaboration and partnership with parent; (8) reflection coaching (9) non-judgmental feedback; (10) conversation and reciprocity; (11) ethical conduct; (12) organization and management of session; and (13) management of conflict or implementation difficulties.

**Interactions with community EI programming.** PIXI is not meant to replace community-based EI programming. PIXI interventionists will encourage all families to connect with EI providers and will make appropriate referrals where warranted. With parental permission, the PIXI interventionist will communicate directly with the family's EI coordinator and collaborate with services providers to support synergistic intervention delivery.

**Project website.** A website and accompanying smartphone app will be available to help sharing information, scheduling sessions, and completing study measures.

**Retention.** Interventionists will be in close contact with all families across their first year of participation. Communication will occur via email and over video-conferencing weekly. A separate team of assessors will communicate with the family to schedule assessments. Resources and access to interventionists will be provided via the project website and app in addition to traditional email/phone call options. Upon the completion of each assessment timepoint, parents will be provided $50 via e-gift card for their time and insight. Parents do not receive financial incentives for participating in intervention sessions.

## Outcomes

Fig 1 summarizes the schedule of outcome measures to be completed, based on the SPIRIT template.

**Primary outcome measures.** Primary outcome measures address feasibility and acceptability of the intervention.

1. Recruitment will be deemed feasible if we can meet our target of 20–30 families enrolled within the study period. Qualitative notes on barriers to participation for those who express interest but cannot enroll will inform potential future challenges for scaling up the intervention model.

2. Retention goals will be assessed by determining the number and percentage of families who complete at least 75% of the proposed 20 sessions and at least two post-intervention assessments. A retention of 80% of the sample (at least 16–24 families completing the sessions) will be considered acceptable.

3. Acceptability of the intervention will be assessed through use of a social validity questionnaire and an interview with each family by an independent research associate not on the

intervention team. The social validity measure was developed by one of the current investigators to examine acceptability of EI programs. The measure asks questions about the family's experience with various components (e.g., materials, activities, interventionist, virtual format) and with the intervention as a whole. This will be conducted after each phase of the intervention. The interview targets domains of acceptance, satisfaction, and feasibility of the intervention as a whole and will be conducted at the conclusion of the intervention. Rapid qualitative analysis of the targeted domains will be conducted following the completion of all participant interviews and will be used to assess acceptability and identify areas for modification/improvement for future iterations of the program.

4. Parent and therapist fidelity of intervention administration will be assessed with the adapted *Parent Implementation Rating Form* [19] and the *Infant Start Parent Fidelity Tool* [19, 20].

The *Parent Implementation Rating Form* includes eight items rated on a 7-point Likert scale by the interventionist after each session. The measure captures parent's participation and engagement behaviors, including (1) readiness for session; (2) use of knowledge and skills between sessions; (3) reflection of changes in child's behaviors; (4) engagement and attentiveness; (5) interactions with infant; (6) relevant questions and comments; (7) demonstrated understanding of topic; and (8) collaboration in generating ideas for practice. Scores were tallied, and an overall rating of effectiveness (minimal, moderate, maximum) was identified for each session.

**Secondary outcome measures.**   The secondary outcome measures examine fidelity of parent implementation, early developmental skills, and autism symptomology using a combination of parent report and observational standardized assessment measures for remote administration.

A modified version of the *Infant Start Parent Fidelity Tool* will be used. This tool includes 14 items rated on a 3-point Likert scale (seldom present, sometimes present, always present). Items are associated with the parenting behaviors and strategies discussed during weekly sessions (see Table 1). Parent fidelity will be consensus-coded based on a review of video-taped interactions for all participants by two team members who are blinded to topic session and number.

A diagnostic clinical assessment will be conducted with each child participant, administered by licensed clinicians who are not part of the intervention team. Participants will receive a remotely delivered assessment that includes a diagnostic interview and the *TELE-ASD-PEDS* [21]. The TELE-ASD-PEDS is a telehealth-based assessment that involves the parent engaging in basic tasks with their young child while being observed by a trained clinician. Observations are combined with diagnostic interview data to determine the likelihood that the child has an ASD. Seven symptoms of ASD are coded on a three-point Likert scale (symptom not present; symptom present but at subclinical levels; symptom obviously consistent with autism). These symptoms are totaled to provide an overall score along with the clinician's certainty of their diagnostic impression.

Parent report of symptoms will be collected through the *Communication and Symbolic Behavior Scales* (CSBS; [22]); *Modified Checklist for Autism in Toddlers* (MCHAT; [23]); the *Repetitive Behavior Scale for Early Childhood* (RBS-EC; [24]); *Sensory Profile-2*, [25] and *Vineland Adaptive Behavior Scales, Third Edition* [26] The CSBS provides an overview of early predictors of language and social communicative behaviors, whereas the MCHAT is a validated tool for screening risk for ASD in children between 16 and 36 months of age. The RBS-EC and Sensory Profile-2 measure repetitive behaviors and sensory issues often observed in children with NGC, ASD, or both. The *Vineland Adaptive Behavior Scales, Third Edition* (Vineland-3;

[26]) is a standardized parent/caregiver report measure of language, motor, social/leisure, and overall adaptive functioning in individuals across the lifespan, measuring what an individual does independently in day-to-day life.

Additional assessments will be collected at each timepoint to provide a comprehensive understanding of infant development within the first 3 years of life for children identified with NGC pre-symptomatically. These assessments include an examination of sleep [27], feeding [28, 29], and behavior [30, 31].

Understanding that an early diagnosis of an NGC can be difficult for parents and a supportive intervention may have positive impacts on caregivers, we will also assess parental mental health at each timepoint. Depressive symptomology/risk will be captured by the *Edinburgh Postnatal Depression Scale* [32] within the first year of life and *Patient Health Questionnaire-9* [33], and symptoms of anxiety will be captured by the *State-Trait Anxiety Inventory* [34]. Parenting stress and social support will be examined using the *Parenting Stress Inventory*, *Fourth Edition*: *Short Form* [35] and *Medical Outcomes Survey*, *Social Support Scale* [36]. Additionally, the *Behavior Rating Inventory of Executive Functioning*, *Second Edition*, *Adult Version* [37] will be completed by participating mothers to capture general insight into cognitive functioning.

**Data management and quality control.** This non-randomized feasibility pilot includes a variety of participant- and family-level data collected through parent report and direct assessment. Questionnaires will be completed primarily through online assessment portals and through the Research Electronic Data Capture system [38]. Data will be collected and managed by PIXI research staff, led by team members not engaged in direct intervention provision. Social validity and parent acceptability (i.e., completion interview) will be completed via phone in an interview format with participants by a qualitative research expert blind to study procedures and participant engagement.

**Statistical methods.** Our primary outcomes are related to feasibility and acceptability metrics, which will involve descriptive data on the number of participants who complete most of the sessions and qualitative review of parent reports of satisfaction with the program as a whole and with specific components. Secondary analysis will involve data pertaining to child development and parent well-being. Individual scores on the secondary data measures will be plotted graphically, and visual analysis will be used to assess change across timepoints. We will also compare the overall mean scores on secondary outcome measures from our sample to that of prior published scores from natural history studies in the target NGCs. Because we will not be powered to accurately interpret results from statistical modeling, we will rely on visual analysis and simple descriptive statistics to inform decisions regarding signals of efficacy to support a larger-scale trial.

**Research ethics approval, protocol amendments, consent.** Institutional Review Board (IRB) approval has been obtained through participating institutions; PIXI is also a registered clinical trial with clinicaltrials.gov (NCT03836300). Prior to participation, a research assistant will review the consent form with potential participants, providing details of what study involvement pertains, including potential benefits and risks. Any changes to the study protocol will undergo review by the IRB of record. Participants will be informed that participation is optional, that they can receive EI services through their community providers, and that they can discontinue with PIXI at any time.

**Confidentiality.** All data will be kept in password-protected share drives or within electronically secure networks. Hard-copy data contain unique ID numbers and will be stored in a locked cabinet separate from identifiers.

**Ancillary and post-trial care.** Participants will be provided contact information for their interventionist, as well as the study coordinator, study PI, and IRB representative. Parent mental health will be monitored formally through regular assessments and informally during study

visits. Materials to support parental coping, including postpartum depression, will be provided to all families, with referrals to specialists made when warranted. The PIXI team includes three licensed mental health professionals who will be available to support interventionists in managing parents in need.

**Dissemination policy.**   The results of this non-randomized feasibility pilot will be published in a peer-review journal, regardless of the outcome of the study. We will also disseminate findings, including barriers and successes at national and international conferences targeting professionals who work with this population.

## Discussion

The primary purpose of this paper is to provide a detailed protocol for a pilot study to assess feasibility, acceptability, and signals of efficacy of participation in an EI program targeted to infants with NGCs and their parents. To our knowledge, this is the only EI program to be developed and tested with very young children with the NGCs being targeted in this program. Efforts to reduce the age at diagnosis and identify newborns or infants before symptoms appear are making headway, especially as prenatal and newborn screening efforts expand. In the absence of disease-modifying therapeutics, it is imperative that we have empirically based psychosocial and behavioral interventions available to support families in reducing symptoms and improving long-term outcomes for their children. Even if efforts to modify the biological course of the conditions are successful, families will still require support in coping and adapting to the diagnosis and making informed decisions about their child's treatment options.

PIXI was developed in response to a need to offer high-quality, empirically based targeted intervention for newborns with NGCs who are identified with their condition through participation in Early Check, a pilot newborn screening study in North Carolina [12]. Given its virtual nature, enrollment will be open to any infant in the United States with a documented diagnosis of the target NGC. We believe offering tailored EI to newly diagnosed families will have long-term impact by setting parents up to be successful and feel competent in supporting their child's development and advocating for their needs. PIXI also provides concrete strategies, along with coaching by an experienced interventionist, to address emerging symptoms that occur during the first year of life.

PIXI is designed to supplement, not replace, community-based EI services. We will monitor use of EI and, where appropriate, work with the EI coordinator for a family to support synergy in goals and strategies across the two programs. Should PIXI ultimately be a successful intervention model for young children with NGCs, our immediate next steps will be to establish training modules and oversight for scalable implementation by EI providers.

Although initially designed for infants with one of five target NGCs, we believe the structure and components proposed for PIXI are applicable and could be beneficial for infants diagnosed with any number of conditions where developmental and behavioral challenges are a key phenotypic feature. This non-randomized feasibility pilot will provide critical information about feasibility, acceptability, fidelity metrics, and efficacy that will set the stage for a potentially larger and more scalable trial.

## Supporting information

**S1 Checklist. SPIRIT 2013 checklist: Recommended items to address in a clinical trial protocol and related documents\*.**
(DOC)

**S1 File.**
(PDF)

**S2 File.**
(PDF)

**S1 Appendix.**
(DOCX)

## Author Contributions

**Conceptualization:** Anne C. Wheeler, Katherine C. Okoniewski, Lauren Turner-Brown.

**Data curation:** Anne C. Wheeler, Samantha Scott, Anne Edwards.

**Funding acquisition:** Anne C. Wheeler, Lauren Turner-Brown.

**Investigation:** Katherine C. Okoniewski, Lauren Turner-Brown.

**Methodology:** Anne C. Wheeler.

**Project administration:** Anne C. Wheeler, Katherine C. Okoniewski, Samantha Scott, Anne Edwards, Emily Cheves.

**Supervision:** Anne C. Wheeler, Katherine C. Okoniewski, Lauren Turner-Brown.

**Validation:** Samantha Scott, Lauren Turner-Brown.

**Visualization:** Anne C. Wheeler.

**Writing – original draft:** Anne C. Wheeler, Katherine C. Okoniewski, Samantha Scott, Anne Edwards.

**Writing – review & editing:** Anne C. Wheeler, Katherine C. Okoniewski, Samantha Scott, Anne Edwards, Emily Cheves, Lauren Turner-Brown.

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
