## [Decision Letter · Decision Letter 0]

14 Sep 2022

PONE-D-22-15391Pilot Protocol for the Parent and Infant Inter(X)action Intervention (PIXI) Feasibility StudyPLOS ONE

Dear Dr. Wheeler,

Thank you for submitting your manuscript to PLOS ONE. After careful consideration, we feel that it has merit but does not fully meet PLOS ONE’s publication criteria as it currently stands. Therefore, we invite you to submit a revised version of the manuscript that addresses the points raised during the review process.

We look forward to receiving your revised manuscript.

Kind regards,

Joseph Donlan

Staff Editor

PLOS ONE

Journal Requirements:

Reviewers' comments:

Reviewer's Responses to Questions

**Comments to the Author**

1. Does the manuscript provide a valid rationale for the proposed study, with clearly identified and justified research questions?

Reviewer #1: Yes

2. Is the protocol technically sound and planned in a manner that will lead to a meaningful outcome and allow testing the stated hypotheses?

Reviewer #1: Yes

3. Is the methodology feasible and described in sufficient detail to allow the work to be replicable?

Reviewer #1: Yes

4. Have the authors described where all data underlying the findings will be made available when the study is complete?

Reviewer #1: No

5. Is the manuscript presented in an intelligible fashion and written in standard English?

Reviewer #1: Yes

6. Review Comments to the Author

You may also provide optional suggestions and comments to authors that they might find helpful in planning their study.

Reviewer #1: Thank you very much for enabling me to review this interesting protocol for a future study – the he Parent and Infant Inter(X)action Intervention (PIXI) Feasibility Study.

It is an exciting study and intervention that the team are planning to undertake and it will be helpful to have this protocol published in order that others who are working in this space can be aware of the study and its methodologies.

There are a number of points that I have highlighted that I hope will be helpful to the authors in revising their paper for publication.

Firstly, in the introduction, the authors provide some background to the present situation where many children are now being diagnosed with genetic syndromes and conditions much earlier and therefore may benefit from early intervention. The authors do not describe at all, the heterogeneity of the genetic conditions that they may well be working with – this will be fundamental to how the intervention is taken on and how a future randomised controlled trial might be considered. It might be important to discuss this a little more in the introduction. For example, some children with William’s syndrome may have few or no long term difficulties besides mild learning difficulties whereas others may be much more severe. On the other hand, some children with some genetic conditions may be degenerative, metabolic conditions or those that require many more health interventions e.g. tracheostomies, gastrostomies, cardiac support, seizures. There is no discussion of the very different nature of these conditions and how this will be taken into account in the trial but also in the feasibility study.

The authors call the study; “a case series” for understanding feasibility of an intervention. I am not clear why it needs to be called a case series. Many pilot and feasibility studies prior to running a trial will have approximately 30 participants and are not necessarily called “case series” unless the authors feel they are doing more specific case work on each case that merits this title?

There are many assumptions in the article and study – in particular, that there is enough of an evidence base that the different modalities that are likely to be affected in children with genetic conditions, warrant intervention and that there is enough evidence that we will see change. I am not sure if this is entirely the case.

The team chose two main intervention packages; Parents as Teachers and Infant Smart. These both seem excellent interventions and I was excited to read about them. From a scientific perspective however, it was not clear from reading the article as to how these had been chosen in a review and whether the authors just knew of these specific programmes and/or whether they had done a scoping or systematic review of the literature. Either way, they look like good programmes – but being really clear about this could be helpful- did the authors do a full review or not – ok if not, but please say so. There are other packages out there that there are not any others that target similar domains, are parent-mediated and have empirical evidence of efficacy – in particular there may be some used in LMIC settings.

The authors describe that they are using a “mixed methods” for undertaking the feasibility study. Are they using any framework such as the MRC Framework for Complex Interventions to undertake this feasibility study?

It was not clear exactly outcomes of the pilot feasibility and acceptability trial will inform the trial and what were there to just inform a revision and recreation of the tool. It feels that these are different things and might be better separated a little more.

The authors describe utilising the SPIRIT framework for the reporting of their protocol. As this is a pilot/feasibility study, would the authors not consider using guidelines on what needs to be reported on and measured in a feasibility and acceptability study e.g. https://pilotfeasibilitystudies.biomedcentral.com/articles/10.1186/s40814-019-0499-1

Objectives

In looking at recruitment rates, if this is a pilot/feasibility study for a future RCT (which is what is mentioned by the authors) – does the team also need to pilot and consider how they will do randomisation? And what inclusion and exclusion factors will be taken into account when recruiting?

In the secondary objectives, the authors describe using outcome measures to assess efficacy of the intervention. In a feasibility study, might it be better at this stage to understand whether they may be valid measures of the outcomes that are expected in the study or whether there are other measures that might need to be considered?

Intervention

It would be helpful to understand who and what training the interventionalists have and whether the team aim to understand or gain any data on the feasibility of training.

It may also be worthwhile to consider how the team will measure cost of the intervention and whether they would be able to take any of this into account when scaling up. So important when it may be possible to demonstrate the cost-benefit of this intervention in comparison to others which may be more expensive when done in person – or the case may be the opposite – but important to consider.

Participants:

Will the team gain any knowledge about the health of the participants in the study? What about children with more complex needs - those with gastrostomies or with severe reflux or cardiac problems? Will these children be included or excluded? Will this information be taken into account or not in a future trial and possibly taken into consideration as a co-variate? Or would it be better to exclude certain children with more complex needs? How will this be considered?

Small point:

Why EPDS and PHQ-9? Many only use one of these measures.

7. PLOS authors have the option to publish the peer review history of their article (what does this mean?). If published, this will include your full peer review and any attached files.

Reviewer #1: No

---

## [Author Response · Author response to Decision Letter 0]

23 Mar 2023

Reviewer #1: Thank you very much for enabling me to review this interesting protocol for a future study – the he Parent and Infant Inter(X)action Intervention (PIXI) Feasibility Study. It is an exciting study and intervention that the team are planning to undertake and it will be helpful to have this protocol published in order that others who are working in this space can be aware of the study and its methodologies. There are a number of points that I have highlighted that I hope will be helpful to the authors in revising their paper for publication.

1. Firstly, in the introduction, the authors provide some background to the present situation where many children are now being diagnosed with genetic syndromes and conditions much earlier and therefore may benefit from early intervention. The authors do not describe at all, the heterogeneity of the genetic conditions that they may well be working with – this will be fundamental to how the intervention is taken on and how a future randomized controlled trial might be considered. It might be important to discuss this a little more in the introduction. For example, some children with William’s syndrome may have few or no long-term difficulties besides mild learning difficulties whereas others may be much more severe. On the other hand, some children with some genetic conditions may be degenerative, metabolic conditions or those that require many more health interventions e.g. tracheostomies, gastrostomies, cardiac support, seizures. There is no discussion of the very different nature of these conditions and how this will be taken into account in the trial but also in the feasibility study.

- This is a great point, and we appreciate the attention drawn to this conceptualization. To address this note we have further delineated that the conditions we will be working with will be ones that are genetic in nature and result in a primary phenotype of intellectual/cognitive/developmental disability to be considered for eligibility. 

2. The authors call the study; “a case series” for understanding feasibility of an intervention. I am not clear why it needs to be called a case series. Many pilot and feasibility studies prior to running a trial will have approximately 30 participants and are not necessarily called “case series” unless the authors feel they are doing more specific case work on each case that merits this title?

- Thank you for this suggestion. In response, we have changed the nomenclature to “non-randomized feasibility pilot” study to be clearer about the purpose of the study

3. There are many assumptions in the article and study – in particular, that there is enough of an evidence base that the different modalities that are likely to be affected in children with genetic conditions, warrant intervention and that there is enough evidence that we will see change. I am not sure if this is entirely the case.

- Thank you for raising this concern. To address this comment, we reviewed the manuscript and reduced language around assumptions about impact and outcomes. There is strong evidence that intervening early in kids with developmental delays (e.g., language, motor, learning, social communication), meeting their needs through zones of proximal development in addition to having an informed and empowered parent can increase competency in support for intervention implementation and future advocacy for needs, which was the aim of the assumptions presented in the introduction. 

4. The team chose two main intervention packages; Parents as Teachers and Infant Smart. These both seem excellent interventions and I was excited to read about them. From a scientific perspective however, it was not clear from reading the article as to how these had been chosen in a review and whether the authors just knew of these specific programmes and/or whether they had done a scoping or systematic review of the literature. Either way, they look like good programmes – but being really clear about this could be helpful- did the authors do a full review or not – ok if not, but please say so. There are other packages out there that there are not any others that target similar domains, are parent-mediated and have empirical evidence of efficacy – in particular there may be some used in LMIC settings.

- We noted in the introduction on page 5 that a functional review (i.e., not a structured/formal comprehensive or scoping review) was completed and the decision was made to include these two programs. 

5. The authors describe that they are using a “mixed methods” for undertaking the feasibility study. Are they using any framework such as the MRC Framework for Complex Interventions to undertake this feasibility study?

- No formal framework will be used within the feasibility study. The mixed methodology approach will include combined qualitative and quantitative procedures to understand the early experiences of families of children with NGCs, response to intervention, and acceptability of the protocol. 

6. It was not clear exactly outcomes of the pilot feasibility and acceptability trial will inform the trial and what were there to just inform a revision and recreation of the tool. It feels that these are different things and might be better separated a little more.

- We aim to do both with this trial. Our initial goals are to identify what changes need to be made globally, and which may be needed for different conditions. We’ll also be examining if the overall model is feasible and acceptable. This information will help us to implement reversions and determine applicability within a larger trial. 

7. The authors describe utilising the SPIRIT framework for the reporting of their protocol. As this is a pilot/feasibility study, would the authors not consider using guidelines on what needs to be reported on and measured in a feasibility and acceptability study e.g. 

Thank you for this suggestion. We have incorporated the CONSORT extension guidelines for pilot trials into this version of the manuscript. 

 

Objectives

In looking at recruitment rates, if this is a pilot/feasibility study for a future RCT (which is what is mentioned by the authors) – does the team also need to pilot and consider how they will do randomisation? And what inclusion and exclusion factors will be taken into account when recruiting?

- We will be considering randomization as data comes in during the trial. We will not pilot any randomization initially as a main goal is to understand basic acceptability and feasibility of the intervention overall. We have added detail regarding inclusion/exclusion factors to our participant and sample size section. 

In the secondary objectives, the authors describe using outcome measures to assess efficacy of the intervention. In a feasibility study, might it be better at this stage to understand whether they may be valid measures of the outcomes that are expected in the study or whether there are other measures that might need to be considered?

- Yes, we entirely agree. Addressed on page 7. 

Intervention

It would be helpful to understand who and what training the interventionalists have and whether the team aim to understand or gain any data on the feasibility of training.

- Added more specific information on licensing of interventionists. We will develop and adapt training materials to align with changes made to the methods of the intervention as iterative changes are made and monitor the fidelity of new interventionists to help assess the feasibility of training. These will be future activities based on the results of the pilot. 

It may also be worthwhile to consider how the team will measure cost of the intervention and whether they would be able to take any of this into account when scaling up. So important when it may be possible to demonstrate the cost-benefit of this intervention in comparison to others which may be more expensive when done in person – or the case may be the opposite – but important to consider.

- Across the trial we will closely be monitoring the cost for implementation inclusive of materials, interventionist time, and training. Addressed on page 9. 

Participants

Will the team gain any knowledge about the health of the participants in the study? What about children with more complex needs - those with gastrostomies or with severe reflux or cardiac problems? Will these children be included or excluded? Will this information be taken into account or not in a future trial and possibly taken into consideration as a co-variate? Or would it be better to exclude certain children with more complex needs? How will this be considered?

- One of the quantitative report measures will ask all parents to report on the concerns and needs of their child. We also ask about feeding and medical needs. For the trial, inclusion determination for those with complex health needs will be made on a case-by-case basis and used to inform RCT inclusion criteria. 

Small point:

Why EPDS and PHQ-9? Many only use one of these measures.

- We use these measures at different timepoints. The EPDS in the first year to allign with public health practices to facilitate referrals for parents as necessary specific to the postpartum period, and the PHQ for the timepoints beyond.

---

## [Editor Report · Decision Letter 1]

19 Apr 2023

Pilot Protocol for the Parent and Infant Inter(X)action Intervention (PIXI) Feasibility Study

PONE-D-22-15391R1

Dear Dr. Wheeler,

We’re pleased to inform you that your manuscript has been judged scientifically suitable for publication and will be formally accepted for publication once it meets all outstanding technical requirements.

Kind regards,

Joseph Donlan

Staff Editor

PLOS ONE
---

## [Editor Report · Acceptance letter]

24 Apr 2023

PONE-D-22-15391R1 

Pilot Protocol for the Parent and Infant Inter(X)action Intervention (PIXI) Feasibility Study 

Dear Dr. Wheeler:

I'm pleased to inform you that your manuscript has been deemed suitable for publication in PLOS ONE. Congratulations! Your manuscript is now with our production department. 

Kind regards, 

on behalf of

Dr Joseph Donlan 

Staff Editor

PLOS ONE